# The Effects of Climate Change on the Distribution Pattern of Species Richness of Endemic Wetland Plants in the Qinghai-Tibet Plateau

**DOI:** 10.3390/plants13141886

**Published:** 2024-07-09

**Authors:** Yigang Li, Danzeng Zhaxi, Ling Yuan, Anming Li, Jianhua Li, Jinhu Wang, Xing Liu, Yixuan Liu

**Affiliations:** 1Hubei Key Laboratory of Quality Control of Characteristic Fruits and Vegetables, College of Life Sciences and Technology, Hubei Engineering University, Xiaogan 432000, China; liyigang0103@163.com (Y.L.); yuanling0614@163.com (L.Y.); lsahmp68@yahoo.com.cn (A.L.); wyzmkm@sohu.com (J.L.); 2School of Ecology and Environment, Tibet University, Lhasa 850000, China; xiaoyaozi5188@sina.com (J.W.); xingliu@whu.edu.cn (X.L.); 3Agriculture and Animal Husbandry Comprehensive Service Center, Jiangrang Township, Cuoqin County, Ngari 859000, China; 15708079555@163.com; 4College of Life Science, Wuhan University, Wuhan 430072, China; 5Key Laboratory of Biodiversity and Environment on the Qinghai-Tibetan Plateau, Ministry of Education, Tibet University, Lhasa 850000, China

**Keywords:** climate change, wetland plant, species loss, species turnover, species gain, priority conservation areas, Qinghai-Tibet Plateau

## Abstract

Wetland ecosystems in the Qinghai-Tibet Plateau (QTP), the region with the richest biodiversity and the most important ecological barrier function at high altitudes, are highly sensitive to global change, and wetland plants, which are important indicators of wetland ecosystem structure and function, are also threatened by wetland degradation. Therefore, a comprehensive study of changes in the geographical distribution pattern of plant diversity, as well as species loss and turnover of wetlands in the QTP in the context of global climate change is of great importance for the conservation and restoration of wetland ecosystems in the QTP. In this study, species turnover and loss of 395 endemic wetland plants of the QTP were predicted based on the SSP2-4.5 climate change scenarios. The results showed that there were interspecific differences in the effects of climate change on the potential distribution of species, and that most endemic wetland plants would experience range contraction. Under the climate change scenarios, the loss of suitable wetland plant habitat is expected to occur mainly in parts of the southern, north-central and north-western parts of the plateau, while the gain is mainly concentrated in parts of the western Sichuan Plateau, the Qilian Mountains, the Three Rivers Source Region and the northern Tibetan Plateau. Overlaying the analysis of priority protected areas with the established protected areas in the QTP has resulted in the following conservation gaps: the eastern Himalayan region, midstream of the Yarlung Zangbo River, the transition zone between the northern Tibetan Plateau and the Hengduan Mountains, Minshan-Qionglai mountain, Anyemaqen Mountains (southeast) to Bayankala (southeast) mountains, the southern foothills of the Qilian Mountains and the northern Tibetan Plateau region. In the future, the study of wetland plant diversity in the QTP and the optimisation of protected areas should focus on the conservation gaps. This study is of great importance for the study and conservation of wetland plant diversity in the QTP, and also provides a scientific basis for predicting the response of wetland plants to climate change in the QTP.

## 1. Introduction

At the macro-scale, climate is considered to be one of the most important drivers of species geographic distributions [1]. Climate change can affect community structure, species geographic distribution patterns, population genetic diversity and ecosystem functions [2]. Global climate change may lead to range expansions and contractions for most species. Meanwhile, habitat loss and fragmentation may also lead to the extinction of threatened species with narrow natural ranges [2,3]. Under future climate change, plants face three outcomes: first, shifts in their current ranges and migration to new suitable areas; second, adaptation to new climatic conditions; and third, facing extinction [4]. Species range shifts are the most directly visible adaptation strategy of species to climate change and have attracted much attention [5]. In response to climate change, many species may migrate to colder habitats to find suitable climatic niches, such as higher latitudes and altitudes, resulting in range shifts [6]. For example, Liang et al. modelled the potential distribution of 151 plant species in the Hengduan Mountains, and the results showed an overall trend of species shifting to higher altitudes and latitudes, with plants in high-altitude regions expanding their range [7].

Predicting species loss and turnover under future climate change has become a highly researched topic in ecology [3,8,9,10]. Under climate change, suitable habitats for species may shift, potentially leading to the disappearance of existing suitable habitats and the emergence of new ones. This phenomenon is known as species habitat loss and gain, and the ratio of these changes represents species turnover. Species habitat loss reflects the extent to which species are threatened by climate change, while habitat gain reflects the extent to which species benefit from climate change. Species habitat turnover, on the other hand, represents their ability to adapt to climate change [8]. Li et al. conducted a study on the turnover and loss of 111 woody plant species in China and found that species in arid and monsoon regions will experience greater species loss under climate change, while species in alpine regions will experience the opposite trend [9]. In the context of climate change, regions with high species turnover rates are often those where population changes are most significant [11]. Therefore, identifying critical habitats for wetland plant conservation in the QTP under climate change requires a focus on areas with high turnover rates. At the same time, climate change is causing species range shifts, and species redistribution poses serious challenges to the static boundaries of existing protected area networks [10,12]. In response to climate change, many species may migrate out of current protected areas, weakening the protective effect of these areas [10]. In addition, climate change may lead to the introduction of new species into protected areas, affecting the conservation objectives and management efforts of existing protected areas [10,13]. Therefore, predicting the effects of climate change on the geographical patterns and turnover rates of wetland plants in the QTP can help managers formulate targeted conservation strategies, further improve the effectiveness of the protected area network and reduce the risk of extinction of wetland plants in the QTP under rapid climate change [10].

Climate change has a profound impact on freshwater ecosystems because aquatic ecosystems and their biodiversity are directly dependent on climatic conditions [14]. Numerous studies have shown that climate warming can affect freshwater ecosystems, leading to changes in the distribution of river fish [15] and macroinvertebrates [16], as well as a decline in global amphibian populations [17]. Climate change may also affect the hydrological conditions of wetlands [18], leading to changes in runoff and flood patterns due to variations in precipitation, glacier melt and an increase in the frequency of extreme events [19]. This may affect the growth of wetland plants, whose life cycles are often closely linked to hydrological conditions. In addition, precipitation variability, prolonged droughts and long-term water stagnation may degrade the quality of freshwater ecosystems such as wetlands, potentially leading to local extinctions of certain species [20]. Makki et al. simulated the current and future spatial distribution of 131 river fish species at 1481 sites in Iran and found that the potential habitat occupancy of 37 species is expected to decline, with the southern Caspian region of Iran experiencing the greatest species decline in the future [21].

The effects of climate change are more pronounced at high altitudes than at low altitudes [5]. Studies have shown that the rate of warming at high altitudes is higher than the global average [5], and this has been confirmed in various mountain ecosystems around the world, such as the Rocky Mountains [22], the European Alps [23], mountain ranges in the western United States [24] and the Great Alpine region [25]. In recent decades, the rate of warming in the QTP has exceeded that of other regions of the northern hemisphere at the same latitude [6]. Alpine ecosystems are particularly sensitive to climate change because alpine biota are often restricted by low temperatures [5]. The low long-distance dispersal ability of alpine plants makes it difficult for them to migrate long distances under climate change [5]. Previous studies have shown that climate change has a significant impact on species composition and distribution in the Tibetan Plateau region [26]. Climate change will create more challenges or opportunities for endemic and threatened species on the Tibetan Plateau [26]. Therefore, studying the effects of climate change on the distribution patterns of endemic wetland plant diversity in the Tibetan Plateau is crucial for the effective conservation of wetland plant diversity in this region.

This study aims to investigate the effects of climate change on the patterns of endemic wetland plant diversity, identify priority conservation areas for endemic wetland plants and assess the conservation effectiveness of current nature reserves for wetland plant priority conservation areas. Specifically, this study aims to (1) predict the potential impacts of climate change on the geographical patterns, species turnover rate, species loss rate and species gain rate of endemic wetland plants in the QTP; (2) identify key areas for wetland plant conservation under climate change scenarios; (3) evaluate the conservation effectiveness of existing nature reserves for wetland plant priority conservation areas and propose targeted conservation recommendations.

## 2. Materials and Methods

### 2.1. Study Area and Species Distribution Data

The topography of the QTP is extremely complex, with dense high mountains separated by deep valleys, resulting in diverse climatic conditions across the plateau. From the southeast to the northwest, temperature and precipitation gradually decrease, and the climate changes from warm and humid to cold and dry [27]. The QTP has complex and diverse plant species and complex sources of plant components, making it an important hotspot of global plant diversity. The wetland ecosystems of the QTP support a rich biodiversity at high altitudes and provide critical ecological barrier functions. They are highly sensitive to global change and are seen as a symbol of global change [28]. We used the urban land and urbanisation index on the Tibetan Plateau (2018, 2019), which was downloaded from the National Tibetan Plateau Data Center (https://data.tpdc.ac.cn/, accessed on 18 January 2024), to determine the geographical area of the QTP [29]. A checklist of 1958 wetland plant species, including 100 aquatic plants and 1858 wetland plants, was compiled using a large number of floras, monographs, online databases, literature and field survey data (Appendix A). The species distribution data of wetland plants reported by Li et al. [30] were used as the main data source in this study. We compared the checklist of endemic seed plants of the QTP published by Yu et al. [31] with the species distribution database of wetland plants of the QTP constructed in this study to determine the list of endemic wetland plants of the QTP. The total number of endemic wetland angiosperms in the QTP was 395, belonging to 39 families and 93 genera.

### 2.2. Environmental Data for Species Distribution Models

The environmental data used for species distribution modelling in this study were classified into climate data, topographic data and land use data. The SSP2-4.5 common socio-economic pathways from the Scenario Model Intercomparison Project (ScenarioMIP) for CMIP6 were selected for the 2050 and 2070 climate scenarios. The SSP2-4.5 selected for this study represent the medium social vulnerability and medium forcing scenarios, respectively, where radiative forcing stabilises at 4.5 W/m^2^ in 2100. Future climate data were downloaded from the Worldclim site (https://www.worldclim.org, accessed on 28 January 2024) with a spatial resolution of 2.5 arcmin, coordinate system GCS_WGS_1984 and model BCC-CSM2-MR from our National Climate Centre. To avoid multicollinearity, Pearson correlation analysis was used to exclude variables with Pearson r > 0.8. Topographic data were divided into elevation and slope. Digital elevation model (DEM, m) data with an accuracy of 250 m were downloaded from the Resource and Environment Science Data Platform (https://www.resdc.cn/, accessed on 18 January 2024). The slope map of the Tibetan Plateau (2000) was downloaded from the National Tibetan Plateau Data Centre (https://data.tpdc.ac.cn/, accessed on 18 January 2024) and slope data with a spatial resolution of 90 m were extracted.

### 2.3. Building Species Distribution Models

Biomod2 is an ensemble species distribution prediction model that has been widely used to predict species distributions across geographic regions and taxa [32]. In this study, the R package “biomod2” [33] was used to model the potential range of wetland plants under future climate change scenarios. To ensure the predictive power of the species distribution model, 500 pseudo-missing points were randomly generated to participate in the modelling. The species distribution record points and species pseudo-missing points were integrated, and 75% of them were selected as the training dataset to calibrate the model, while the remaining 25% were used to evaluate the model performance, and the process was repeated three times [32]. In this study, the performance of the different base models was assessed using the True Skill Statistic (TSS), where the closer the value of this metric is to 1, the better the performance of the model [34]. Only the base models with a TSS ≥ 0.7 were retained for the construction of the combined model using the weighted average method [34]. Using the natural breakpoint method to classify the results, we obtained a gradient classification of suitable habitat divided into four levels: 0~0.05 representing unsuitable habitat; 0.05~0.33 representing marginally unsuitable areas; 0.33~0.66 representing marginally suitable areas, and values greater than 0.66 indicating the most suitable areas. In this study, a probability of occurrence of 0.66 or greater was selected as a potentially suitable area for wetland plants [35].

### 2.4. Calculating Species Turnover, Loss and Gain under Climate Change Scenarios

Under climate change scenarios, the loss of species habitats can reflect the level of threat to species. By calculating the turnover rate of wetland plants, we can better predict their adaptability to future climate change (e.g., migration in or out) [8]. The species is absent in the current climate scenario but present in the future climate scenario, indicating an expansion of its potential range in that grid cell; (3) persistence in the grid cell: the species is present in both the current and future climate scenarios; (4) absence: the species is absent in both the current and future climate scenarios [36]. In the context of widespread species migration, the formulae for calculating the turnover rate, loss rate and gain rate of wetland plant species in each grid cell are as follows [8,9,37]:$$\;T=\frac{L\;+G}{\mathrm{SR}\;+\;G}$$
$$\mathrm{SL}=\frac{L}{\mathrm{SR}}$$
$$\mathrm{SG}=\frac{G}{\mathrm{SR}}$$
where T is the species turnover rate, SL is the species loss rate and SG is the species gain rate. L represents the number of species lost within a grid cell under future climate scenarios (equivalent to the number of species that emigrate) and G represents the number of species gained within a grid cell under future climate scenarios (equivalent to the number of species that immigrate). SR represents species richness under the current climate scenario. Species turnover (T) is commonly used to measure the intensity of species change within a grid cell. A higher species turnover rate indicates greater changes in species composition within that grid cell [10].

### 2.5. Identification of Priority Conservation Areas and Conservation Gaps

Based on the richness of endemic wetland plant species, changes in species richness, species loss and species gain, priority conservation areas are identified under future climate change scenarios. These areas are grouped into three main categories: (1) “Areas requiring protection and attention”: These areas have concentrated wetland plant distributions with high species richness, which in particular remain stable even under the influence of climate change. These areas are less vulnerable to climate change and may serve as important refugia for endemic wetland plants under climate change, and therefore require significant protection and attention. (2) “Areas requiring protection and restoration”: These areas have high rates of species loss and significant declines in species richness. These areas are fragile and highly vulnerable to climate change, making them priority areas for future conservation and restoration of endemic wetland plants. (3) “Areas requiring protection and exploration”: These areas have increasing trends in wetland plant species richness and high rates of species acquisition. These areas are likely to provide a significant amount of suitable habitat for endemic wetland plants in the future, and therefore require significant conservation and research.

This study identifies the “Areas requiring protection and attention”, “Areas requiring protection and restoration” and “Areas requiring protection and exploration” under future climate change scenarios as priority areas for wetland plant conservation. To assess the conservation status of these priority conservation areas, digital maps of nature reserves in the QTP are overlaid with the maps of priority conservation areas to identify conservation gaps in protection. Data for nature reserves in the QTP were obtained from the dataset published by Liu et al. [38]. For each grid cell (covering an area of 2500 km^2^), if there was no nature reserve in the region, it was designated as a “conservation gap” [39]. To provide a more accurate quantification of priority conservation areas in the QTP, all geographical pattern analyses in this chapter were based on 50 km × 50 km grid units.

## 3. Results

### 3.1. Impact of Climate Change on the Potential Distribution of Endemic Wetland Plants in the QTP

The distribution patterns of unique wetland plants in the QTP predicted by the model under current climate conditions are generally consistent with the actual distribution ranges of these plants obtained from specimen and survey data (Figure 1). Under the influence of climate change, the species richness of unique wetland plants overall maintains a relatively stable geographical distribution pattern, mainly concentrated in the eastern Himalayas and the Hengduan Mountains region. However, there are still some changes in species richness in specific local areas. In particular, species richness in the Aba region of the western Sichuan Plateau is projected to increase in 2050 and 2070, with a clear upward trend. In contrast, the southern part of Chamdo City in south-eastern Tibet and some areas adjacent to Shannan City and Nyingchi City are projected to show a decreasing trend in species richness (Figure 1).

The potential impact of climate change on species distributions varies from species to species. The results indicate that the majority of endemic wetland plants will experience range contraction under climate change (Appendix A). Specifically, it is predicted that by 2050 and 2070, the distribution areas of 130 and 121 endemic wetland plants will decrease by more than 20%, respectively, while the distribution areas of 95 and 90 endemic wetland plants will decrease by more than 40%, respectively. However, the results also show that a very small number of endemic wetland plants will benefit from climate change by increasing their range (Appendix A). Of these, 12 and 18 endemic wetland plant species are predicted to increase their range by more than 20% by 2050 and 2070, respectively (Figure 2).

### 3.2. Impact of Climate Change on the Migration of Endemic Wetland Plants in the QTP

Under the climate change scenario, there is little difference in average species turnover, average species loss and average species gain across all grid cells between 2050 and 2070. The average species loss is higher than the average species gain in both periods (Table 1). The results show that the spatial distribution of species turnover, species loss and species gain is inhomogeneous. Among them, the geographical patterns of species turnover and species loss are relatively consistent. The turnover and loss of wetland plant habitats mainly occur in parts of the southern, north-central and north-western regions of the plateau. Habitat gain is mainly in the western Sichuan Plateau, the Qilian Mountains and some parts of the Three Rivers Source and the northern Tibetan Plateau (Figure 3).

### 3.3. Key Areas for the Protection of Endemic Wetland Plants in the QTP under Climate Change

In this study, the “Areas requiring protection and attention” are mainly located in the Qionglai-Daxueshan-Sharuli Mountains, the Taniantaweng Mountains, parts of the central and southern parts of the Hengduan Mountains and some regions in the eastern Himalayas. These regions serve as core habitats for unique wetland plants, maintaining high species richness under current and future climate conditions. They represent one of the most critical conservation areas for endemic wetland plants. On the other hand, the “Areas requiring protection and restoration” are mainly located in the middle reaches of the Yarlung Zangbo river basin and some parts of the eastern Himalayas. These areas are fragile and highly vulnerable to the effects of climate change. In addition, the “Areas requiring protection and restoration” are mainly located in the southeastern part of the Bayankala Mountains, the Minshan Mountains, the southern foothills of the Qilian Mountains and some parts of the northern Tibet Plateau. These regions show an increasing trend in wetland plant species richness and may provide many suitable habitats in the future (Figure 4).

Combining the overlap between priority protected areas and nature reserves, the results show that there are still a large number of priority conservation areas that are not covered by protected areas (Figure 4), and that the conservation gaps are mainly in the eastern Himalayan region, the middle Yarlung Tsangpo River valley, the transition zone between the northern Tibetan Plateau and the Hengduan Mountains, Minshan-Qionglai mountain, the Anyemaqen Mountains (southeast), Bayankala (southeast) mountain, the southern foothills of the Qilian Mountains, and the northern Tibetan Plateau region (Figure 5).

## 4. Discussion

### 4.1. Effects of Climate Change on the Distribution Patterns of Endemic Wetland Plants in the QTP

The results of this study show that both the actual distribution areas of endemic wetland plants on the Tibetan Plateau and the potential distribution areas under current and future climate scenarios predicted by species distribution models are mainly concentrated in the Himalayan and Hengduan mountain regions. The results suggest that mountainous regions play a crucial role in maintaining wetland plant diversity under the influence of climate change. For example, hotspots for species richness of gymnosperms and oak (*Quercus* L.) species are also located in mountainous areas under climate change in China [40,41]. In addition, the Karkonosze Mountains in Poland serve as an important refugium for plant communities in the mountainous regions of central Europe [42]. The complex terrain, large altitudinal gradients, high environmental heterogeneity and relatively low human disturbance in mountain regions provide suitable habitats and refugia for various species [42]. In addition, terrain heterogeneity has a buffering effect on climate warming. It mitigates the impact of climate change on species survival by increasing the availability of suitable habitats within short distances, allowing species to persist locally during periods of continuous climate change [9,43,44]. Therefore, these regions have the potential to provide a significant number of suitable habitats for endemic wetland plants of the QTP in the future.

The effects of climate change on the distribution of alpine species do not follow a fixed pattern, but vary according to the life history and resource requirements of the species [5]. Due to differences in ecological characteristics between species, the ranges of narrowly distributed species may continue to narrow, while those of widely distributed species may expand [45]. The results indicate that the potential range of most endemic wetland plants will continue to shrink, and a few species may even become threatened with extinction. At the same time, a small number of endemic wetland plants will benefit from climate change and may expand their range in the future. This suggests that the impact of climate change will vary between different endemic wetland species. Li et al. studied the range changes of 208 endemic and endangered species in China under climate change and found that 50% of the species will experience range contraction, while the remaining 50% will experience range expansion [46]. Yousefi et al. studied the effects of climate change on the current and future ranges of 15 endemic freshwater fish species in Iran and found that 5 species will lose some of their current suitable habitats, while 10 species will gain new suitable habitats [47]. The above results show that the impact of climate change varies between taxa, confirming that the influence of climate change on species distributions does not follow a fixed pattern. It is generally accepted that species with smaller ranges under current climate conditions are more likely to be at risk of local extinction under climate change scenarios [48,49]. This is because their restricted distribution is often taken as an indicator of low ecological tolerance and high sensitivity to environmental change [5]. In future research, we will focus on the differences in response to climate change between broad-spectrum and narrow-spectrum wetland plants in the QTP, and explore the general laws of these two types of plants in dealing with climate change.

### 4.2. Patterns and Explanations of Species Turnover, Loss and Gain under Climate Change

The southern region of the QTP has permafrost, with abundant recharge from snowmelt from modern glaciers. There are extensive swamps, mainly herbaceous swamps. In contrast, the north-western plateau region is characterised by an arid and semi-arid climate, with low annual rainfall and insufficient moisture, which is not conducive to the formation and development of wetland vegetation. Wetland vegetation types are simple, with only herbaceous marshes, mainly shallow vegetated wetlands and inland salt marshes [50]. Wetland vegetation in these areas is relatively homogeneous, and the proportion of endemic wetland plants is relatively low. Therefore, the loss of a small amount of suitable habitat will result in a higher rate of species loss, potentially leading to the degradation or collapse of local wetland ecosystems. The regions with higher species gain of endemic wetland plants in the QTP are mainly located in the western Sichuan Plateau region in the northern Hengduan Mountains and the Qilian Mountains region in the north-eastern plateau. These regions will provide a large number of suitable habitats for endemic wetland plants in the QTP in the future. For example, under future climate change scenarios, the potential range of threatened conifers endemic to China is mainly concentrated in the western Sichuan Plateau [41]. The Hengduan Mountains have long been regarded as a refuge for plants under harsh climatic conditions [7]. The Hengduan Mountains are mainly composed of mountains and rivers running from north to south, serving as a corridor for plants to migrate along altitudes and latitudes in response to climate change [6]. During past climate change, the Hengduan Mountains region provided shelter for species and short dispersal distances to favourable habitats [43,51]. At the same time, the steep ecological gradients in this region may also buffer the effects of regional climate variability by promoting species range shifts, enabling plants to withstand climate change [6,43]. During the Quaternary glaciation, the northward route from the northern part of the Hengduan Mountains to the Qilian Mountains served as one of the critical corridors for species dispersal in the Tibetan Plateau region [52]. The complex terrain of the Hengduan Mountains has effectively buffered the effects of past climate change and may continue to play a similar role in future climate change, thereby promoting the survival of species [43]. Under future climate change scenarios, species on the Tibetan Plateau may continue to migrate along this corridor [52]. Liang et al. found that the Hengduan Mountains region may support higher species diversity under future climate change than what currently exists [7]. The complex terrain of the Hengduan Mountains region may buffer the effects of future climate change on endangered alpine endemic species. Therefore, under future climate change, the Hengduan Mountains and Qilian Mountains regions may become climatic refugia for endemic wetland plants unique to the QTP. The results also suggest that some parts of the Qiangtang Plateau and the Three-River Source may also serve as climatic refugia for endemic wetland plants under future climate change. For example, Wang et al. found that in addition to the eastern Himalayas and the Hengduan Mountains, the central region of the QTP is also a potential climatic refugium for Meconopsis species [43]. In addition, previous studies have shown that micro-refuge areas also existed in the Yarlung Zangbo River Basin and the Three-River Source region during the Quaternary glaciation [52].

### 4.3. Suggestions for Future Conservation of Wetland Plants in the QTP

Our research has shown that most of the priority conservation areas for endemic wetland plants in the QTP are within nature reserves. This is crucial for the conservation of wetland plant diversity and also reflects the effectiveness of biodiversity conservation efforts in China. Taking the Hengduan Mountain region as an example, several national nature reserves have been established in this area, including Ruoergai Wetland National Nature Reserve, Haizishan National Nature Reserve, Yading National Nature Reserve, Gongga Mountain National Nature Reserve in western Sichuan and Baimaxueshan National Nature Reserve in northwestern Yunnan Province. The establishment of these reserves can effectively protect the endemic wetland plants of the Tibetan Plateau under both current and future climate change scenarios. However, there are still some conservation gaps that need to be addressed. The conservation gaps are mainly located in the eastern Himalayan region, the middle reaches of the Yarlung Zangbo River, the transition zone between the northern Tibetan Plateau and the Hengduan Mountains, the Minshan-Qionglai Mountains, the Anyemaqen Mountains (southeast)-Bayankala Mountains (southeast), the southern foothills of the Qilian Mountains and the northern Tibetan Plateau region. Therefore, it is suggested that the current biodiversity conservation system can be appropriately optimised to improve the conservation efficiency of plant diversity in the wetlands of the Tibetan Plateau.

Under climate change scenarios, endemic wetland plants in the QTP may face even greater losses of suitable ranges or species extinctions than the results of this study suggest. This study assumes that species distributions are primarily influenced by factors such as climate, terrain and land use, and that species are free to migrate and colonise all suitable climatic zones. However, other biological factors such as evolutionary history, species dispersal ability and interspecific interactions may also constrain species ranges [6]. At the same time, the tracking of suitable habitats by species faces many challenges, including terrain complexity, species dispersal ability, longevity, interspecific interactions and the absence of dispersal vectors [53,54]. For example, the complex terrain in the south-eastern part of the Tibetan Plateau provides suitable habitats for wetland plants, but also presents obstacles to the migration and dispersal of many species [55]. The dependence of wetland plants on aquatic environments may further limit their ability to disperse, making it difficult for them to find suitable climates. Karuno et al. studied the effects of climate change on 11 endemic amphibian and reptile species in the QTP and found that some species with limited geographical ranges distributed along river valleys, such as *Phrynocephalus theobaldi*, *Cyrtodactylus tibetanus* and *Cyrtopodion medogensis*, may face challenges in finding suitable climates [44]. Therefore, there is uncertainty as to whether endemic wetland plants will be able to keep pace with future climate change. In addition, in this study, endemic wetland plants with fewer than five distribution points in the QTP were excluded from the modelling analysis due to their relatively narrow distribution ranges, resulting in a limited number of available specimen records. If these narrowly distributed endemic species were included in the analysis, the proportion of endemic species with shrinking ranges would increase further [7]. Therefore, in future conservation planning, it is necessary to continuously monitor and pay attention to those endemic wetland species with a narrow distribution range to ensure their effective protection. Given the uncertainty of carbon dioxide emission scenarios and the theoretical limitations of the methods used in this study, the results of this study cannot be predicted with complete accuracy [9,56]. Despite the uncertainties and limitations, the preliminary assessment of the potential range of endemic wetland plants in the QTP remains one of the most effective methods available. This chapter first evaluates the impact of climate change on endemic wetland plants in the QTP, and the research results can provide a reference for the formulation of wetland plant conservation policies in the QTP. Future studies need to conduct more surveys based on the wetlands of the QTP and include more comprehensive factors to more accurately predict the distribution of endemic wetland plants in the QTP.

## 5. Conclusions

This study focuses on the conservation of wetland plant diversity in the QTP, predicting the effects of future climate change on the distribution patterns of endemic wetland plants, identifying critical areas for the conservation of endemic wetland plants under future climate change scenarios, and assessing the conservation effectiveness of current nature reserves in protecting the critical areas. The results showed that there were interspecific differences in the effects of climate change on the potential distribution of species, and that most endemic wetland plants would experience range contraction under climate change. Under the climate change scenario, the loss of suitable habitat for wetland plants is expected to occur mainly in parts of the southern, north-central and north-western plateau, while the gain is mainly concentrated in parts of the western Sichuan Plateau, the Qilian Mountains, the Three Rivers Source Region and the northern Tibetan Plateau. By overlaying the priority conservation areas with existing nature reserves in the QTP, the following conservation gaps have been identified: the eastern Himalayan region, midstream of the Yarlung Zangbo River, the transition zone between the northern Tibetan Plateau and the Hengduan Mountains, Minshan-Qionglai mountains, Anyemaqen Mountains (southeast) to Bayankala (southeast) mountains, the southern foothills of the Qilian Mountains and the northern Tibetan Plateau region. In the future, the survey of wetland plant diversity in the QTP and the optimisation of reserves should focus on conservation gaps.

## Figures and Tables

**Figure 1 plants-13-01886-f001:**
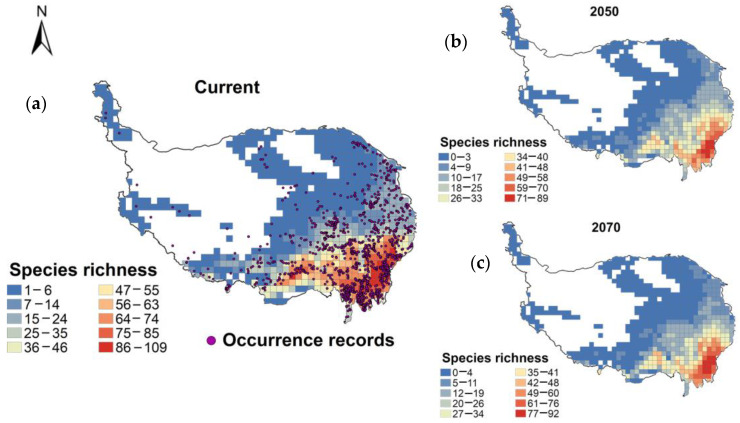
Geographical distribution patterns of endemic wetland plants species richness on the Qinghai-Tibet Plateau under current and future climate conditions, based on 50 km × 50 km grids. (**a**) Current distribution pattern; (**b**,**c**) distribution patterns in 2050 and 2070.

**Figure 2 plants-13-01886-f002:**
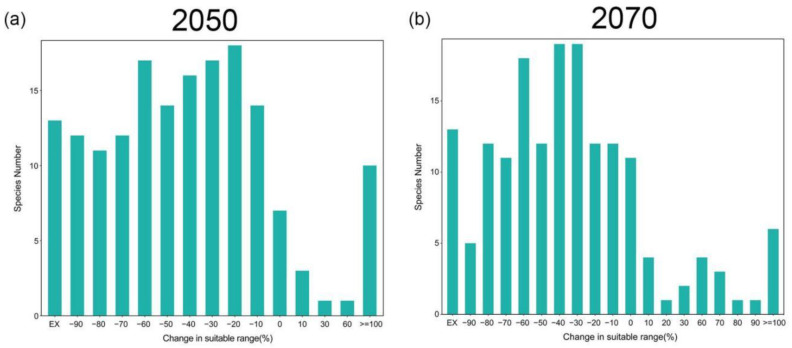
Percentage change in suitable habitat under climate change for endemic wetland plants in the QTP in 2050 and 2070. Negative numbers on the *x*-axis indicate a decrease in habitat.

**Figure 3 plants-13-01886-f003:**
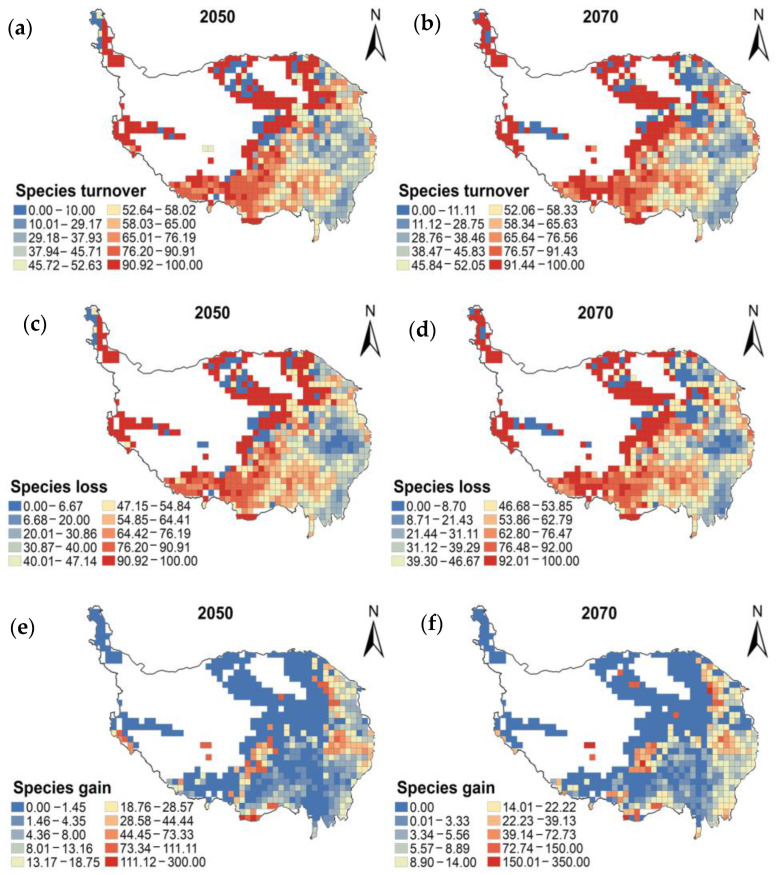
Distribution patterns of species turnover (**a**,**b**), species gain (**c**,**d**) and species loss (**e**,**f**) of endemic wetland plants on the Qinghai-Tibet Plateau in 2050 and 2070, based on 50 km × 50 km grids.

**Figure 4 plants-13-01886-f004:**
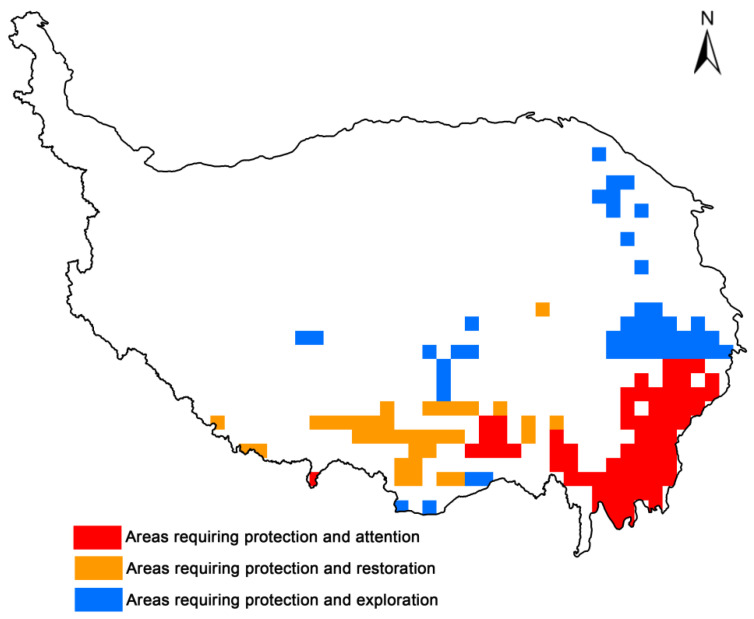
“Areas requiring protection and attention” (red), “Areas requiring protection and restoration” (yellow) and “Areas requiring protection and exploration” (blue) under future climate change scenarios.

**Figure 5 plants-13-01886-f005:**
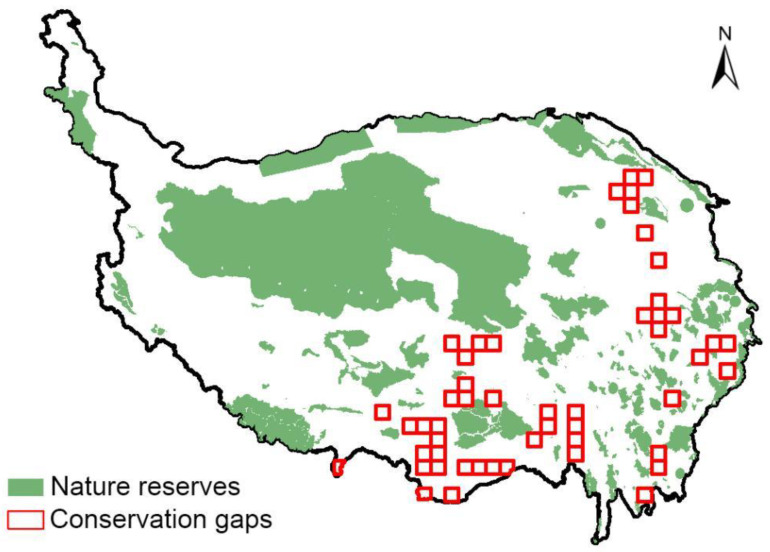
Geographical distribution of conservation gaps for wetland plants in the Qinghai-Tibet Plateau under future climate change scenarios. The red grid represents conservation gaps and the green colour represents nature reserves.

**Table 1 plants-13-01886-t001:** Statistical results of average percentages of wetland species loss, gain and turnover in 2050 and 2070 under climate change scenarios.

	2050	2070
Species loss	59.01%	58.44%
Species gain	11.30%	10.20%
Species turnover	62.43%	61.57%

## Data Availability

Datasets are available upon request to the authors.

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
