# Peer review of "The Effects of Climate Change on the Distribution Pattern of Species Richness of Endemic Wetland Plants in the Qinghai-Tibet Plateau"

_plants, 2024, doi:10.3390/plants13141886_

Round 1
Reviewer 1 Report
Comments and Suggestions for Authors
Interesting article that deals with the richness of endemic species in wetlands, which are endangered by climate change, since their distribution area is affected. According to the authors, this is causing habitat loss and the appearance of new habitats.
The introduction and methodology sections are correct. The results speak of areas in which the model predicts an increase in species and other areas in which there will be a decrease in a projection to 2050-2070. In the discussion they talk about different mountainous areas affected by climate change and specifically about Quercus species.
According to the authors, some wetland species will be favored and will expand their distribution area, but most of them will have more restricted areas.
In general, the work is well structured, being a study of interest to be published and to be known by the scientific community, for which I congratulate the authors.
However, in order to improve the study, I suggest that the authors incorporate in the results a table with the scientific name of the endemic species that will have restrictions in their area and those that will experience expansion. If there is a restriction of area, there may be a loss of genomes, and it is important that this is known by the administrations and that they can remedy the situation.
Author Response
Reviewer #1: Interesting article that deals with the richness of endemic species in wetlands, which are endangered by climate change, since their distribution area is affected. According to the authors, this is causing habitat loss and the appearance of new habitats.
The introduction and methodology sections are correct. The results speak of areas in which the model predicts an increase in species and other areas in which there will be a decrease in a projection to 2050-2070. In the discussion they talk about different mountainous areas affected by climate change and specifically about Quercus species.
According to the authors, some wetland species will be favored and will expand their distribution area, but most of them will have more restricted areas.
In general, the work is well structured, being a study of interest to be published and to be known by the scientific community, for which I congratulate the authors.
However, in order to improve the study, I suggest that the authors incorporate in the results a table with the scientific name of the endemic species that will have restrictions in their area and those that will experience expansion. If there is a restriction of area, there may be a loss of genomes, and it is important that this is known by the administrations and that they can remedy the situation.
Response: Thank you for your kind approval of our work. We sincerely appreciate the thorough comments provided by the reviewer. The reviewer's suggestion is indeed very important. It can put our research into practice and provide an important reference for the conservation of wetland plant diversity in the Qinghai-Tibet Plateau. We have made the necessary additions in accordance with the reviewer's requests, which are detailed in Supplementary Table 2 and 3. We sincerely hope that our response will meet the expectations of the reviewers.

Reviewer 2 Report
Comments and Suggestions for Authors
The article presents a comprehensive study on the impact of climate change on the distribution of endemic wetland plants in the Qinghai-Tibet Plateau (QTP), a critical biodiversity hotspot.The manuscript is well structured and easy to follow. Nevertheless some issues should be addressed. In particular:
1. Use passive voice within the text.
2. Leave spaces before references and the last word before the reference.
3. You should provide better analysis images for the maps. Also, enlarge them. At this point are not readable.
4. Provide more detailed information about the geographical and ecological characteristics of the Qinghai-Tibet Plateau.
5. Include a brief description of why various data sources and their respective databases were chosen.
6. While the use of the R package “biomod2” is mentioned, it would be helpful to explain why this particular package was chosen over others.
7. Include more details on how TSS values were interpreted and their thresholds for model performance would be useful.
8. Explain the rationale behind generating 500 pseudo-missing points for the SDMs.
9. Include a comparative analysis of how the findings from QTP align or differ from the other mentioned studies. This is the meaning of discussion.
Comments on the Quality of English Language
A minor editing of the English language is required.
Author Response
Reviewer #2: The article presents a comprehensive study on the impact of climate change on the distribution of endemic wetland plants in the Qinghai-Tibet Plateau (QTP), a critical biodiversity hotspot.The manuscript is well structured and easy to follow. Nevertheless some issues should be addressed. In particular:
Response: We appreciated the reviewer’s positive and encouraging feedbacks, and thank for your valuable comments. We have made careful revisions based on your comments, and the specific revisions and responses are as follows:
- Use passive voice within the text.
Response: Thank you very much for your useful comments. In the earlier manuscripts we missed this important point. We have revised it in line with your comments.
- Leave spaces before references and the last word before the reference.
Response: Thank you for pointing out this error. We have corrected this error in the revised manuscript.
- You should provide better analysis images for the maps. Also, enlarge them. At this point are not readable.
Response: Thanks for pointing this out and the detailed description. We have revised Figure 4 and redrawn it according to your comments (page 8 and 9, lines 297- 305). We hope the effect of the image meets your requirements.
- Provide more detailed information about the geographical and ecological characteristics of the Qinghai-Tibet Plateau.
Response: We appreciate the reviewer's insightful suggestion. As the reviewer pointed out, the previous manuscript lacked a description of the physical characteristics of the Qinghai-Tibet Plateau and focused only on the sources of environmental factors. We have corrected this error in the revised manuscript (page 3, lines 128-136).
- Include a brief description of why various data sources and their respective databases were chosen.
Response: Thanks for pointing this out and we had corrected as suggested. We have attached the source of the database construction in S1, please check it.
- While the use of the R package “biomod2” is mentioned, it would be helpful to explain why this particular package was chosen over others.
Response: Thanks for pointing this out and we had corrected as suggested.
- Include more details on how TSS values were interpreted and their thresholds for model performance would be useful.
Response: Thank you for this thoughtful and detailed comment, we have made changes in line with the reviewer's comments.
- Explain the rationale behind generating 500 pseudo-missing points for the SDMs.
Response: We appreciate the reviewer's insightful suggestion. The main reasons for choosing 500 pseudo-presence points are as follows: Firstly, 500 pseudo-presence points can adequately cover the study area of the Tibetan Plateau. Secondly, 500 points is a relatively reasonable number, and increasing the number of pseudo-presence points does not improve the accuracy of the model predictions.
- Include a comparative analysis of how the findings from QTP align or differ from the other mentioned studies. This is the meaning of discussion.
Response: Thank you for pointing this out. As you mentioned, comparing the differences in different outcomes may better reflect the value of the discussion. We have already changed the text accordingly (page 10, lines 339-341 and lines 345-348).

Round 2
Reviewer 2 Report
Comments and Suggestions for Authors
Thank you for addressing my comments.